# Selective Preference Optimization via Token-Level Reward Function Estimation

## Abstract

Recent advancements in large language model alignment leverage token-level supervisions to perform fine-grained preference optimization. However, existing token-level alignment methods either optimize on all available tokens, which can be noisy and inefficient, or perform selective training with complex and expensive key token selection strategies. In this work, we propose Selective Preference Optimization (SePO), a novel selective alignment strategy that centers on efficient key token selection without requiring strong, fine-grained supervision signals. We theoretically prove the feasibility of Direct Preference Optimization (DPO) as token-level reward function estimators, which applies to any existing alignment datasets and enables cost-efficient token selection with small-scale model sizes and training data. We then train an oracle model with DPO on the target data and utilize the estimated reward function to score all tokens within the target dataset, where only the key tokens are selected to supervise the target policy model with a contrastive objective function. Extensive experiments on three public evaluation benchmarks show that SePO significantly outperforms competitive baseline methods by only optimizing on 30% key tokens. We also explore SePO as a new paradigm for weak-to-strong generalization, showing that weak oracle models effectively supervise strong policy models with up to $16.8\times$ more parameters. SePO also selects useful supervision signals from out-of-distribution data, alleviating the over-optimization problem. The project is open-sourced here.

## 1 Introduction

The recent development of large language models (LLMs) has focused on aligning model outputs with human intentions and preferences (Ji et al., 2023). During alignment, LLMs are directly optimized on pairwise data and response-level supervision, where popular methods such as reinforcement learning from human feedback (RLHF) (Ouyang et al., 2022; Stiennon et al., 2020) and direct alignment algorithms (Rafailov et al., 2024c; Yuan et al., 2023; Meng et al., 2024) only introduce supervision signals at the end of each response. As deriving preference optimization as bandit problems can lead to sub-optimal solutions and unstable training processes (Zhong et al., 2024; Zeng et al., 2024), many works propose to model LLM decoding as token-level Markov Decision Processes (MDP) and introduce step-wise supervision signals that quantify the value of each action, successfully applied in tasks such as instruction following (Zhong et al., 2024; Yoon et al., 2024) and mathematical reasoning (Xie et al., 2024; Chen et al., 2024b; Lai et al., 2024).

Though achieving outstanding performance, most of these methods are optimized on all available tokens from the training dataset. To validate the effectiveness of this setup, in Figure 1, we present the token-level reward accumulations for 1,000 samples from an instruction following dataset (Cui et al., 2023) and a question answering (QA) dataset (Wu et al., 2024), where the token-level rewards are assigned by GPT-4 (Achiam et al., 2023). According to the results, a limited number of tokens with extreme reward values (key tokens) dominate the total rewards. For instruction following, the Top-35.6% tokens occupy the highest 80% rewards for chosen responses, while the lowest 37.4% tokens only occupy 20% rewards for rejected responses. In QA, selecting top-50% key tokens can cover over 80% of effective supervision signals. These observations prove that not all tokens are equally effective in preference alignment, and optimizing on all available tokens can be noisy and inefficient (Lin et al., 2024; Chen et al., 2024c). Some works explored only optimizing on selected response fragments, but their selection strategies are complex and expensive, utilizing Monte-Carlo

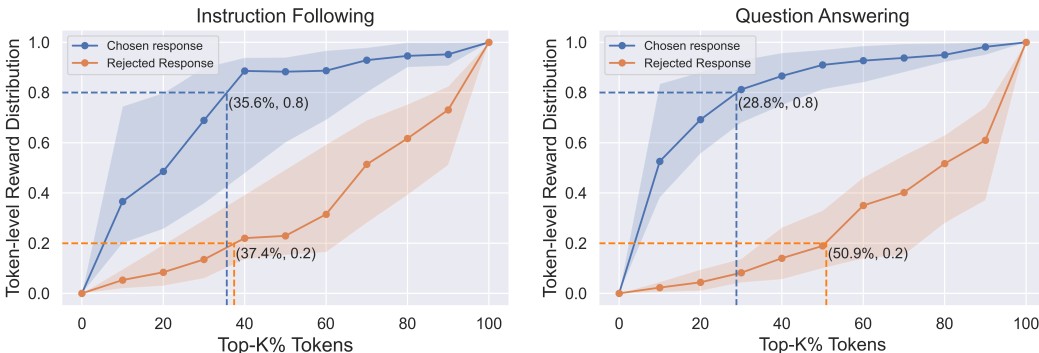

Figure 1: Token-level reward accumulations on two tasks. As tokens with high rewards are considered key tokens for chosen responses, their Top-K% tokens are accumulated in descending order with the highest rewards. In contrast, rewards are accumulated in ascending orders for rejected responses. More details about the figure are presented in Appendix D.

Tree Search (Xie et al., 2024; Chen et al., 2024b) or annotations from human/capable LLMs (Lai et al., 2024; Yoon et al., 2024). The above limitations underscore the importance of selective training and efficient token selection strategies in improving preference optimization algorithms.

Based on these intuitions, we propose Selective Preference Optimization (SePO), a novel selective alignment strategy that centers on efficient key token selection without requiring strong, fine-grained supervision signals. We theoretically show that Direct Preference Optimization (DPO) (Rafailov et al., 2024c) inherently learns a reward function that decouples the response-level reward values into token level (Rafailov et al., 2024b). Based on this conclusion, we propose the first DPO-based token selection method, which trains an oracle model on a moderate-scale subset of the target data distribution, aiming to parameterize an optimal token-level reward function. This token selection strategy has two key advantages: 1) the oracle modeling process is based on the original response-level preference annotations without requiring any extra supervision signals, making it directly applicable to any existing alignment datasets; 2) the cost for token selection can be easily reduced by controlling the oracle model size and the scale of its training subset, which enables cost-efficient selective alignment. We then utilize the estimated reward function to score all tokens within the large-scale target dataset, where tokens with the highest reward values in the chosen responses and tokens with the lowest reward values in the rejected responses are selected as key tokens for alignment. Finally, we design a reference model-free contrastive objective function to optimize the target policy model on the selected tokens.

As SePO enables small oracle models to steer selective alignment for much larger policy models, we further explore it as new paradigm for weak-to-strong generalization (Burns et al., 2023). Instead of leveraging weak models to provide supervision, 1) we leverage weak oracle models to select tokens from in-distribution data for training strong policy models; 2) we propose to train oracle models on out-of-distribution data, which select key tokens to improve target policy model performance and alleviate over-optimization (Gao et al., 2023; Rafailov et al., 2024a) on the weak supervision data.

In summary, our main contributions are:

- We propose SePO, the first DPO-based selective training strategy for preference alignment, which applies to any alignment datasets with response-level supervision signals and enables cost-efficient token selection with small-scale oracle models and training data.

- Exploration on weak-to-strong generalization shows that weak oracle models effectively supervise strong policy models with up to 16.8× more parameters. SePO also selects useful tokens from weak supervision data, alleviating the over-optimization problem on out-of-distribution data;

- We examine SePO on three public evaluation benchmarks. Experimental results show that SePO significantly improves performances on five policy models and outperforms competitive baseline methods by only optimizing 30% key tokens on the target dataset.

Figure 2: The pipeline for Selective Preference Optimization (SePO). SePO mainly consists of three steps: 1) Parameterize a token-level reward function by training a ref-oracle model pair on a moderate-scale dataset; 2) Score all tokens within the target preference dataset and select key tokens; 3) Train the policy model on selected tokens in a contrastive optimization manner.

## 2 PRELIMINARY

**Preference Alignment of Language Models** Preference alignment directly optimizes LLMs with human preferences by maximizing the reward values of model outputs, which are obtained via a response-level reward function $r(q, y)$. The reward function is defined under the Bradley-Terry (Bradley & Terry, 1952) model of preferences. Specifically, for the same prompt $q$ and two completed responses $(y_1, y_2)$ under data distribution $\mathcal{D}$, the model assumes:

$$P_\mathcal{D}(y_1 \succ y_2|q) = \sigma(r(q, y_1) - r(q, y_2)) \tag{1}$$

where $\sigma$ denotes the logistic function and $P_\mathcal{D}(y_1 \succ y_2)$ denotes the probability that $y_1$ is preferred against $y_2$. The alignment of language models is typically cast as a KL-constrained optimization problem on the reward values, formalized as

$$\underset{\pi}{argmax}\, \mathbb{E}_{q\sim\mathcal{D}, y\sim\pi(y|q)}\left[r(q, y)\right] s.t.\, \mathbb{E}_{q\sim\mathcal{D}}\mathbb{D}_{KL}\left[\pi(y|q)\|\pi_{ref}(y|q)\right] \leq \sigma$$

where $\pi$ denotes the aligned policy model, $\pi_{ref}$ denotes the reference policy model. Taking the following Lagrangian, the problem is transformed as:

$$\underset{\pi}{argmax}\, \mathbb{E}_{q\sim\mathcal{D}, y\sim\pi(y|q)}\left[r(q, y)\right] - \beta\mathbb{D}_{KL}\left[\pi_\phi(y|q)\|\pi_{ref}(y|q)\right] \tag{2}$$

**Direct Preference Optimization** Ziebart et al. (2008) have shown that Eqn 2 has a closed-form optimal solution, which enables the reward function to be re-parameterized by the optimal policy:

$$r(q, y) = \beta\, log\, \frac{\pi^*(y|q)}{\pi_{ref}(y|q)} + \beta\, log\, Z(q) \tag{3}$$

where $\pi^*$ denotes the optimal policy, and $Z(q)$ is the partition function. DPO (Rafailov et al., 2024c) bypasses the reward modeling stage by directly substituting this closed-form solution into Eqn. 1, which cancels out $Z(q)$ as it un-changes with the same $q$, yielding the following DPO objective:

$$\mathcal{L}_{DPO} = -\mathbb{E}_{(q, y_w, y_l)\sim\mathcal{D}}\log\sigma\left(\beta\log\frac{\pi_\theta(y_w|x)}{\pi_{ref}(y_w|x)} - \beta\log\frac{\pi_\theta(y_l|x)}{\pi_{ref}(y_l|x)}\right) \tag{4}$$

where $y_w$ and $y_l$ denote the preferred and dis-preferred responses to the prompt $q$.

## 3 METHODOLOGY

In this section, we first show DPO training as inherently fitting a token-level reward function that decouples the response-level reward function (Sec. 3.1). Based on this conclusion, we propose SePO, which optimizes the target policy model only with selected key tokens by the estimated optimal token-level reward function (Sec. 3.2). We also explore SePO as a new paradigm for weak-to-strong generalization (Sec. 3.3). The training pipeline of SePO is shown in Figure 2.

## 3.1 DPO as Token-level Reward Function Estimator

Due to the auto-regressive nature, the decoding process of LLMs can be naturally formulated as a token-level MDP. The MDP is defined as a tuple $\mathcal{M} = (\mathcal{S}, \mathcal{A}, f, r(s_t, a_t), \rho)$, where $\mathcal{S}$ and $\mathcal{A}$ denote the state space and action space. $s_t \in \mathcal{S}$ deontes the current state, consisting of all prompt tokens and current generated tokens (i.e. $s_t = \{q|y_0, ..., y_t\}$, | denotes concatenation). $s_T$ denotes the terminal state. $a_t \in \mathcal{A}$ denotes the current action, where $\mathcal{A}$ is the token vocabulary. $f$ is the deterministic state transition function that concatenates the current state $s_t$ and action $a_t$. $\rho$ is an initial state distribution over prompts $q$, where the initial state $s_1$ consists of the tokens from $q$. $r(s_t, a_t)$ denotes the token-level reward function.

In this section, we show that DPO inherently learns the best estimate on a token-level distribution of the response-level reward values along the token-level MDP trajectory. We begin with the following widely used mild assumption (Rafailov et al., 2024b; Zhong et al., 2024; Zeng et al., 2024):

**Assumption 1.** *Any response-level reward function $r$ can be decoupled into the token level, as a linear combination of reward values modeled by another token-level reward function $\hat{r}$ along the trajectory for its MDP.*

$$r(q, \tau) = \sum_{t=1}^{T} \hat{r}(s_t, a_t) \tag{5}$$

*where $s_t$, $a_t$ are states and actions within the token-level MDP trajectory $\tau = \{s_1, a_1, s_2, ..., s_T\}$.*

With the above assumption, the Bradley-Terry model in Eqn. 1 can be replaced into token level:

$$P_{\mathcal{D}}(\tau^w \succ \tau^l | q) = \sigma(\sum_{i=1}^{N} \hat{r}(s_i^w, a_i^w) - \sum_{j=1}^{M} \hat{r}(s_j^l, a_j^l))$$

where $\tau^w$ and $\tau^l$ are the chosen and rejected trajectories (responses), which are assumed to start and end at the same state $s_1$ and $s_T$.

**Theorem 1.** *With a reference model $\pi_{ref}$, fitting any reward functions $r$ that are consistent to the Bradley-Terry model with the DPO algorithm leads to an optimal estimation of another reward function $\hat{r}$ that decouples the response-level reward values into the token level, which satisfies:*

$$\hat{r}(s_t, a_t) \propto log \frac{\pi^*(a_t|s_t)}{\pi_{ref}(a_t|s_t)} \tag{6}$$

*where $\pi^*$ denotes the oracle model obtained via DPO on the reference model.*

*Proof Sketch.* This proof is inspired by Rafailov et al. (2024b). Starting with a RL objective with an entropy bonus and a KL-divergence regularization, the optimization process aims to maximize the expected cumulative reward. Under the maximum entropy RL setting, the optimal policy $\pi^*$ is related to the Q-function and value function. The Bellman equation incorporates the KL term, relating the optimal Q-function $Q^*$ to the token-level reward $r(s_t, a_t)$.

By combining these relationships, the token-level reward can be expressed as:

$$r(s_t, a_t) = \beta \log \frac{\pi^*(a_t|s_t)}{\pi_{ref}(a_t|s_t)} + V^*(s_t) - V^*(s_{t+1}),$$

where $V^*(s_t)$ is the optimal value function. Under Assumption 1, summing the token-level rewards over all time steps yields the response-level reward. The term $V^*(s_1)$ (the initial state's value) is constant with the same starting state, which does not affect preference comparisons. Therefore, the preference modeling depends only on the sum of the log-ratio terms. This shows that the optimal policy $\pi^*$ inherently aligns an optimal token-level reward function:

$$\hat{r}(s_t, a_t) = \beta \log \frac{\pi^*(a_t|s_t)}{\pi_{ref}(a_t|s_t)},$$

which indicates Eqn. 6 and completes the proof. See Appendix C.1 for a detailed proof. $\square$

This reward function marks the contribution of each action given the current state at the token level. In practice, the quality of the training data for DPO determines how well the calculated reward quantifies the token-level contribution.

## 3.2 SELECTIVE PREFERENCE OPTIMIZATION

SePO is guided by the simple idea that "not all tokens are equally effective", which has been widely evaluated (Lin et al., 2024; Chen et al., 2024b; Lai et al., 2024). We explore fully utilizing DPO to efficiently select the most useful tokens in modeling human preferences in LLM alignment. Firstly, we train an oracle model on a moderate-scale preference dataset with DPO, aiming to model a token-level reward function for the target data distribution. The reward function is then applied to large-scale data to score all the tokens. The policy model is only trained on selected tokens with highest scores in chosen responses and lowest scores in rejected responses, which are expected as key tokens in achieving alignment.

**Oracle Modeling with DPO.** We explore fitting an oracle model with DPO utilizing a moderate-scale dataset via random sampling over the target preference dataset $\mathcal{D}$, which we expect to parameterize a pessimistic estimation of the target reward function.

**Theorem 2.** *Let $\mathcal{D}$ be the target preference dataset with $N$ samples, and $\mathcal{S}$ be a random selection of $m$ samples from $\mathcal{D}$ ($m \leq N$), which is drawn independently and uniformly. The reward function $r_{\mathcal{S}}$ modeled by fitting $\mathcal{S}$ with DPO is a pessimistic estimation of the target reward function $r_{\mathcal{D}}$. The result can be formalized as:*

$$\mathbb{E}_{\mathcal{S}}(r_{\mathcal{S}}(q, y)) \leq r_{\mathcal{D}}(q, y) \tag{7}$$

*where $q, y$ denote any query-response pairs drown from $\mathcal{D}$. The equality holds when $m = N$.*

*Proof Sketch.* As the reward functions are parameterized via fitting the DPO algorithm, we replace Eqn. 3 into Eqn. 7 and reduce this inequality to comparing the expected optimal policy functions:

$$\mathbb{E}_{\mathcal{S}}[\log \pi_{\mathcal{S}}^*(y|q)] \leq \log \pi_{\mathcal{D}}^*(y|q)$$

Since $\mathcal{S}$ is a uniform random sample from $\mathcal{D}$, the empirical distribution $P_{\mathcal{S}}$ is an unbiased estimator of the true distribution $P_{\mathcal{D}}$; that is, $\mathbb{E}_{\mathcal{S}}[P_{\mathcal{S}}(X)] = P_{\mathcal{D}}(X)$. Therefore, training on $\mathcal{S}$ yields an unbiased estimate of the optimal policy: $\mathbb{E}_{\mathcal{S}}[\pi_{\mathcal{S}}^*(y|q)] = \pi_{\mathcal{D}}^*(y|q)$.

Applying Jensen's inequality for the concave logarithm function, we have:

$$\mathbb{E}_{\mathcal{S}}[\log \pi_{\mathcal{S}}^*(y|q)] \leq \log \mathbb{E}_{\mathcal{S}}[\pi_{\mathcal{S}}^*(y|q)] = \log \pi_{\mathcal{D}}^*(y|q)$$

showing that the expected log-optimal policy from $\mathcal{S}$ is less than or equal to that from $\mathcal{D}$ and completes the proof. Equality holds when $m = N$. See Appendix C.2 for a detailed proof. $\square$

With the results from Theorem 2, we first perform SFT on a base model and the chosen responses of the moderate-scale dataset $\mathcal{S}$ to obtain the reference model $\pi_{ref}$:

$$\mathcal{L}_{SFT} = -\mathbb{E}_{(q,y_w) \sim \mathcal{S}} \sum_i \log \pi_{ref}(y_w^i | q, y_w^{<i}) \tag{8}$$

With the reference model, we further perform DPO on $\mathcal{S}$ with the objective function Eqn. 4 to obtain the oracle model $\pi_{ora}$. With Theorem 1, we can utilize $\pi_{ref}$ and the oracle model $\pi_{ora}$ to parameterize the optimal token-level reward function for human preferences, which are used for token selections.

**Token Selection.** We score all tokens within the target preference dataset with the estimated token-level reward function. Based on the token-level MDP and Theorem 1, for each prompt-response pairs $(q, y)$, the score for token $y_i$ is calculated as follows:

$$s(y_i) = \log \frac{\pi_{ora}(y_i | q, y_{<i})}{\pi_{ref}(y_i | q, y_{<i})} \tag{9}$$

We define a token selection ratio $k$, which determines the selected proportion for each response. For chosen responses, we utilize the following indicator function for selection:

$$\mathbf{I}_k^w(s(y_i)) = \begin{cases} 1, & \text{if } s(y_i) \text{ ranks in highest } k\% \text{ in } y \\ 0, & \text{otherwise} \end{cases} \tag{10}$$

For rejected responses, we change the condition for indicating 1 to "if $s(y_i)$ ranks in lowest $k\%$ in $y$" and mark the indicator function as $\mathbf{I}_k^l(s(y_i))$. The intuition behind this action (Rafailov et al., 2024b; Zhong et al., 2024) is that key tokens for chosen responses are likely to contribute high token-level rewards, while key tokens for rejected responses are likely with low token-level rewards, whose probabilities are significantly suppressed in $\pi_{ora}$ compared to the reference model.

**SePO Objective.** We design a simple contrastive preference optimization objective on the target policy model $\pi_t$ with the selected tokens. Specifically, the objective function is designed as follows:

$$\mathcal{L}_{SePO} = -\mathbb{E}_{(q,y_w,y_l)\sim\mathcal{D}} \log \sigma \left( \hat{u}_w(q, y_w, k_w) - \hat{u}_l(q, y_l, k_l) - \lambda \right)$$

$$\hat{u}_w(q, y, k) = \frac{\gamma}{|y| \cdot k\%} \sum_{i=1}^{|y|} \mathbf{I}_k^w(s(y_i)) \log \pi_\theta(y_i|q, y_{<i}) \tag{11}$$

$$\hat{u}_l(q, y, k) = \frac{\gamma}{|y| \cdot k\%} \sum_{i=1}^{|y|} \mathbf{I}_k^l(s(y_i)) \log \pi_\theta(y_i|q, y_{<i})$$

where $\gamma$ controls the scaling of the rewards, $\lambda$ is controls the contrastive margin, and $k_w$, $k_l$ denote the token selection rate for chosen/rejected responses. The SePO objective enables direct adjustment of only crucial tokens for alignment, which avoids noise in both chosen and rejected responses. This design is expected to improve the efficiency and stability of the preference optimization process. The objective is also length-normalized and reference model-free, which prevents bias towards over-length responses and enables memory-efficient alignment (Meng et al., 2024; Yuan et al., 2023).

## 3.3 SePO for Weak-to-Strong Generalization

A unique advantage of SePO is that the cost of its token selection process can be easily reduced by controlling the base model size, which enables small oracle models to steer the alignment for policy models with much stronger capabilities. Therefore, we further explore SePO as a new paradigm for weak-to-strong generalization (Burns et al., 2023), which aims to elicit strong student models with weak supervision signals. Instead of directly leveraging weak models to provide supervision signals, we propose the following two novel methods (illustrated in Appendix Figure 7):

**Weak Oracle Modeling.** We propose to leverage weak oracle models that underperform the target policy models to select the key tokens from in-distribution data. Our intuition is that weak supervisors (oracle models) only identify which tokens are most effective in enhancing the alignment performance, rather than directly providing supervision signals, which normally requires stronger capabilities than the student model.

**Weak Data Supervision.** As the policy model becomes stronger, continual full optimization on the original data distribution can lead to over-optimization on the reward function (Gao et al., 2023; Rafailov et al., 2024a), which can seriously degrade policy model performance. Online preference optimization (Xiong et al., 2024; Xie et al., 2024) alleviates over-optimization with online annotations of new in-distribution data, but can be costly for strong policy models.

Weak data supervision focuses on scenarios when only weak out-of-distribution data is available for strong policy models. We propose to leverage the SePO framework to select key tokens from the weak dataset, and only the selected tokens are utilized to supervise the strong policy model. Instead of full optimization on the training data, we expect selective optimization on the key tokens to prevent over-optimization on the out-of-distribution data, while still leveraging effective supervision signals to further improve the policy model.

## 4 Experiments

This section introduces key experimental settings. More details can be found in Appendix E.

### 4.1 Experimental Settings

**Models and Training Data.** We comprehensively evaluate SePO on two representative model families: LLaMA (Touvron et al., 2023) and Pythia (Biderman et al., 2023). To approximate the optimal token-level reward function, we first obtain the reference models by training on the UltraChat-200K dataset (Ding et al., 2023) in an SFT manner. For the LLaMA model family, we train the reference model on the pre-trained TinyLLaMA-1.1B (Zhang et al., 2024) base model. For Pythia, we separately train on Pythia-(70M, 160M, 410M, 1B, 1.4B) to facilitate research on the effect of oracle models with different sizes. For each reference model, we obtain the oracle models by further

fine-tuning with DPO on the UltraFeedback dataset (Cui et al., 2023). We examine the selective preference training process on target policy models with various capabilities. For Pythia, we perform SFT on Pythia-(2.8B, 6.9B) with UltraChat-200K and use them as target policy models, which we refer as Pythia-SFT. For LLaMA, we test on TinyLLaMA-Chat-1.1B, LLaMA2-Chat-7B, and LLaMA2-Chat-13B.

| Policy Model | Methods | Arena-Hard Win Rate | AlpacaEval 2.0 | | MT-Bench GPT-4o |
|---|---|---|---|---|---|
| | | | LC Win Rate | Win Rate | |
| Pythia-SFT-2.8B | Base | 2.34% | 3.8% | 4.12% | 2.8 |
| | +DPO | 5.71% | 5.72% | 6.1% | 3.16 |
| | +IPO | 5.6% | 4.8% | 4.96% | 3.12 |
| | +RRHF | 4.37% | 4.33% | 4.47% | 2.93 |
| | +SimPO | 5.2% | 5.8% | 6.0% | 3.3 |
| | +TDPO | 6.2% | 6.58% | 6.8% | 3.26 |
| | +SePO-rand | 3.07% | 4.26% | 4.4% | 2.86 |
| | +SePO (Ours) | **6.3%** | **7.1%** | **7.32%** | **3.45** |
| Pythia-SFT-6.9B | Base | 4.23% | 5.0% | 5.17% | 3.58 |
| | +DPO | 10.2% | 12.78% | 13.27% | 4.7 |
| | +IPO | 8.1% | 11.78% | 12.6% | 4.34 |
| | +RRHF | 7.47% | 11.42% | 13.2% | 4.31 |
| | +SimPO | 8.0% | 11.8% | 12.72% | 4.51 |
| | +TDPO | 10.68% | 13.92% | **13.7%** | 4.78 |
| | +SePO-rand | 4.82% | 5.28% | 5.46% | 3.45 |
| | +SePO (Ours) | **10.94%** | **14.27%** | 13.6% | **5.09** |
| Tiny-LLaMA-chat | Base | 1.6% | 1.26% | 1.43% | 3.28 |
| | +DPO | 2.2% | 1.95% | 2.18% | 3.31 |
| | +IPO | 2.3% | 1.85% | 2.06% | 3.38 |
| | +RRHF | 3.1% | 3.02% | 2.12% | 3.4 |
| | +SimPO | 2.61% | 1.3% | **3.7%** | 3.28 |
| | +TDPO | 3.47% | 2.93% | 2.74% | 3.42 |
| | +SePO-rand | 1.52% | 1.26% | 1.57% | 3.26 |
| | +SePO (Ours) | **4.1%** | **3.55%** | 3.37% | **3.78** |
| LLaMA2-Chat-7B | Base | 4.6% | 5.4% | 5.0% | 4.48 |
| | +DPO | 8.5% | 7.8% | 6.71% | 5.43 |
| | +IPO | 8.12% | 8.78% | 9.4% | 5.64 |
| | +RRHF | 9.4% | 13.35% | 14.41% | 5.35 |
| | +SimPO | 9.59% | 13.58% | **15.4%** | 5.63 |
| | +TDPO | 9.23% | 10.86% | 10.7% | 5.55 |
| | +SePO-rand | 6.73% | 6.38% | 6.47% | 5.23 |
| | +SePO (Ours) | **10.3%** | **14.4%** | 14.91% | **6.38** |
| LLaMA2-Chat-13B | Base | 12.0% | 8.4% | 7.7% | 5.7 |
| | +DPO | 13.48% | 13.72% | 13.37% | 5.84 |
| | +IPO | 13.95% | 14.27% | 14.4% | 5.76 |
| | +RRHF | 13.84% | 15.94% | 16.36% | 5.73 |
| | +SimPO | 14.7% | 16.4% | 17.02% | 5.7 |
| | +TDPO | 14.4% | 15.0% | 15.65% | 6.37 |
| | +SePO-rand | 10.05% | 8.16% | 7.5% | 5.66 |
| | +SePO (Ours) | **15.5%** | **17.53%** | **18.41%** | **6.86** |

Table 1: Performance of SePO and other methods on three benchmark datasets. Oracle models are based on TinyLLaMA-1.1B and Pythia-1B, trained on the full UltraFeedback dataset. The reward function is used to select the top-30% tokens on chosen/rejected responses.

**Baseline Methods.** We compare the performance of SePO with state-of-the-art offline preference optimization methods. For response-level alignment methods, we select DPO (Rafailov et al., 2024c), IPO (Azar et al., 2024), RRHF (Yuan et al., 2024) and SimPO (Meng et al., 2024). We also include token-level alignment method TDPO (Zeng et al., 2024) as it does not require fine-grained supervision signals. To evaluate the SePO token selection process, we further include a self-implemented SePO-rand baseline that randomly selects k% tokens from the pair-wise data and optimizes via Eqn. 11.

**Evaluation Benchmarks.** We quantify the contribution of each method by evaluating on three widely utilized instruction-following benchmarks: AlpacaEval 2.0 (Dubois et al., 2024), MT-Bench (Zheng et al., 2024b), and Arena-Hard (Li et al., 2024). AlpacaEval 2.0 consists of 805 queries to evaluate the models' versatile conversational abilities. Following the standard setting, we report win rates and length-controlled (LC) win rates of evaluated models against GPT-4-turbo responses. The LC win rates are designed to reduce influences of model verbosity.

MT-Bench covers eight categories with 80 queries. We report the average scores ranging from 0 to 10. Arena-Hard extends MT-Bench with 500 high-quality queries, where we report win rates against GPT-4-0314 model outputs. All judgments are performed by the latest GPT-4o model.

## 4.2 OVERALL PERFORMANCE

Performances of SePO and other baseline methods on three benchmark datasets are presented in Table 1. According to the results, SePO significantly improves performance over the base policy models, with an average of 4.47% improvement in win rates on Arena-Hard. SePO also outperforms other strong preference optimization methods. On MT-Bench, SePO achieves the best average scores among other methods, surpassing both state-of-the-art response-level methods such as SimPO and token-level method TDPO. As **SePO models are only optimized on 30% of the tokens trained on other methods**, these results directly strengthen the effectiveness of selective training strategies applied in preference optimization. Further comparisons with SePO-rand show that optimizing on randomly selected k% tokens significantly damages the performance of selective training, proving the effectiveness of our DPO-based token selection strategy in filtering the crucial supervision signals from the training data.

On AlpacaEval 2.0, SePO continues to achieve superior performances over baseline methods on both win rates and LC win rates, further proving its effectiveness on different benchmarks. Notably, SePO outperforms all other methods on length-controlled win rates, including SimPO and RRHF, which are specifically designed for length-normalized reward formulation. These results show that selective training strategies also enhance policy models in avoiding over-length responses. We believe the reason is that during token selection, the token-level reward function can assign the end-of-sentence tokens with low-ranking scores, which can be discarded during optimization if the response ends inappropriately (e.g. over-length or repetition). In contrast, though SimPO and RRHF design length-normalized rewards, the end-of-sentence tokens are included for all responses, still fitting the response lengths for all training samples.

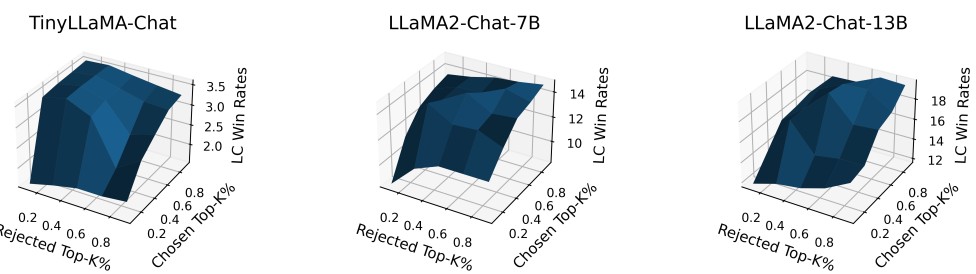

Figure 3: SePO performances with different combinations of K% selection ratios for chosen and rejected responses. The performances are quantified by the LC win rates on AlpacaEval 2.0. All results are obtained with a TinyLLaMA oracle model trained on the full UltraFeedback dataset.

## 4.3 IMPACT OF DATA SCALE

This section evaluates how training data scales of SePO and oracle modeling impact policy model performance. We focus on two research questions:

**How do token selection rates influence SePO performance?** We investigate the influences of token selection rates on SePO performances by introducing various combinations for chosen and rejected responses. The ratio for chosen/rejected responses is each selected from $\{0.1, 0.3, 0.5, 0.7, 0.9\}$ and matched pair-wise, with 25 combinations in total. The experimental results are presented in Figure 3.

According to the results, increasing selection rates from 0.1 for chosen/rejected responses rapidly improves policy model performance, but the momentum decreases as the ratio continues to grow. For example, the LC win rate of LLaMA2-Chat-7B improves from 8.37% to 14.8% as the ratios for chosen/rejected responses rise from 0.1 to 0.5 progressively, then stabilizes around 14.7% with higher selection rates. These observations prove our hypothesis that not all tokens are equally effective for LLM alignment. Training only on key tokens effectively improves alignment performance, while other tokens provide limited supervision information. From the figures, we conclude that training on Top-30% tokens for TinyLLaMA or Top-50% tokens for LLaMA2-Chat-(7B, 13B) already provides comparable performance to aligning on all tokens.

Increasing selection rates for chosen responses generally outperforms increasing ratios for rejected responses. For example, with a fixed rejected selection rate, the TinyLLaMA-Chat performance smoothly improves as the chosen ratios grow. However, improving rejected ratios from 0.1 to 0.9 leads to decreased model performance in 4 out of 5 fixed chosen selection rates. Similar results can be observed in the other two policy models. These results indicate that compared to increasing probabilities for irrelevant tokens in chosen responses, suppressing probabilities for high-reward tokens in rejected responses can significantly affect the model performance.

**How much data trains a good oracle model?** In Theorem 2, we proved that training an oracle model on a moderate-scale subset is a pessimistic estimation of the target reward function. In this section, we empirically investigate the influence of the training data scale ($\frac{m}{N}$ in Theorem 2) for

oracle models. Specifically, we randomly sample different proportions of data from UltraFeedback as training data for the TinyLLaMA-based oracle model. The results are shown in Figure 4.

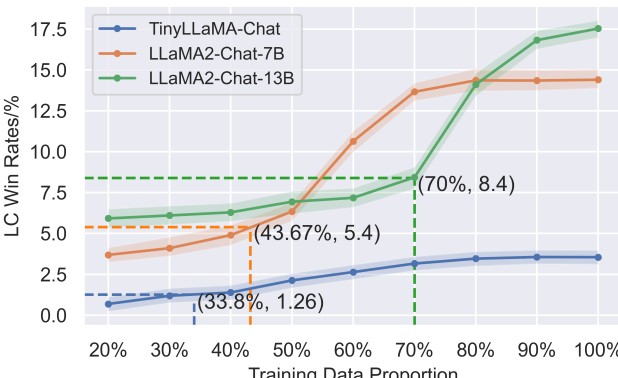

According to the results, training the oracle model on higher proportions of the target data generally leads to superior model performance. LC win rates on all policy models improve as the estimated token-level reward function becomes more accurate. Training on high data proportions also retains the majority of token selection capabilities. For example, supervising LLaMA2-Chat-7B policy model with an oracle model trained on 70% of the data still achieves 13.66% of LC win rates, which outperforms strong baseline methods such as SimPO and RRHF. However, the continual decrease in training proportions can significantly harm model performance. For the

Figure 4: LC win rates on AlpacaEval 2.0, supervised by oracle models trained with different data proportions. We report the average performance of 3 random runs.

TinyLLaMA-Chat policy model, an oracle model trained with less than 40% of target data leads to LC win rates of less than 1.26%, which even underperforms the original policy model. For LLaMA2-Chat-(7B,13B), this threshold increases to 50% and 70%. These results indicate the importance of accurate estimation of the reward function, where false selection of key tokens degrades the capability of the policy model. These thresholds also increase with the size of policy models, showing that the high quality of key tokens becomes more important in supervising models with strong capabilities.

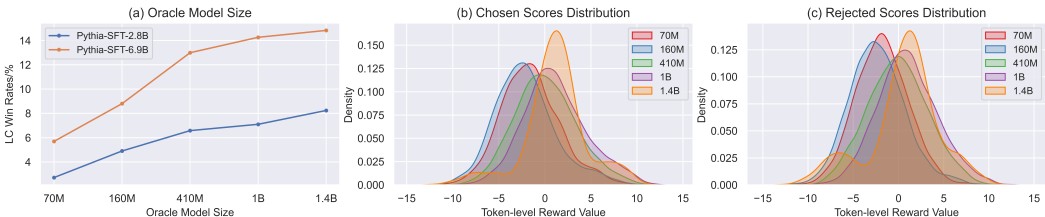

Figure 5: Evaluation results on weak-to-strong generalization. (a) LC win rates of policy models on AlpacaEval 2.0, which are trained with oracle models of various sizes; (b)(c) token-level reward distributions for 5,000 chosen/rejected responses obtained from oracle models with different sizes.

## 4.4 WEAK-TO-STRONG GENERALIZATION

In this section, we empirically evaluate SePO on enhancing weak-to-strong generalization.

**Weak Oracle Modeling.** In Table 1, we presented the performance of TinyLLaMA and Pythia-1B oracle models on guiding stronger policy models (e.g. Pythia-SFT-6.9B, LLaMA2-Chat-13B). The competitive results of SePO prove the viability of weak oracle modeling. To provide a clear landscape, we further train oracle models with Pythia-(70M, 160M, 410M, 1B, 1.4B) on the same target data and compare their performances on the Pythia-SFT-(2.8B, 6.9B) policy models. The results are shown in Figure 5 (a).

According to the results, oracle models with weak capabilities can provide effective supervision to strong policy models. For example, training with the Pythia-410M oracle model achieves 6.58% on Pythia-SFT-2.8B and 13% on Pythia-SFT-6.9B policy models, with up to $16.8\times$ more parameters than the oracle model. These results outperform full optimization on the target dataset with baseline methods such as DPO and SimPO. In addition, the performance of target policy models continually

improves as the oracle model size increases. For example, on Pythia-SFT-6.9B policy model, the 1.4B oracle model outperforms the 410M orcale model by 1.84% and the 70M model by 9.15%. These results show that oracle models with stronger capabilities can better model the token-level reward function and accurately select key tokens.

To provide an intuitive view, we present the token-level score distributions of different oracle models in 5 (b)(c). For both chosen/rejected scores distribution, strong oracle models such as Pythia-(1B,1.4B) show higher densities in extreme (large and small) reward values, which facilitates separating key tokens from the other tokens. In contrast, small oracle models tend to fit a Gaussian distribution, where most tokens have similar scores. These results show that strong oracle models excel in distinguishing key tokens within the dataset, which further proves the capability of the oracle models as crucial in accurately modeling the token-level reward function.

| Methods | Arena-Hard Win Rate | AlpacaEval 2.0 | |
| --- | --- | --- | --- |
| | | LC Win Rate | Win Rate |
| Base | 12.0% | 8.4% | 7.7% |
| +DPO | 10.63% | 7.42% | 7.18% |
| +IPO | 9.5% | 6.5% | 5.98% |
| +RRHF | 11.7% | 7.82% | 7.4% |
| +SimPO | 11.39% | 7.5% | 7.35% |
| +SePO | **13.63%** | **8.81%** | **8.4%** |

Table 2: Performance of SePO and other baseline methods on generalizing the weak HH-RLHF dataset to the strong LLaMA2-Chat-13B policy model.

**Weak Data Supervision.** We evaluate the weak data supervision performance of SePO by training on HH-RLHF (Bai et al., 2022), an early-released preference dataset with relatively lower quality (Yang et al., 2024a) on responses. We perform SePO with a TinyLLaMA-based oracle model and 30% token selection rates, and comparisons with baseline methods are shown in Table 2. According to the results, SePO is the only method that improves the strong LLaMA2-Chat-13B policy model with data from HH-RLHF, outperforming base performance by 1.63% on Arena-Hard and 0.41% on AlpacaEval 2.0. With full optimization, baseline methods such as DPO and SimPO continuously degrade model performance due to over-optimization on weak supervision data. These results prove SePO effective in leveraging useful supervision signals from weak data while avoiding over-fitting harmful patterns. These results point to SePO as a highly efficient method for continually improving strong model performance with large-scale out-of-distribution data.

## 4.5 RELATED WORK

Previous works related to SePO can be divided into three parts: response-level preference optimization (Ouyang et al., 2022; Stiennon et al., 2020; Rafailov et al., 2024c; Yuan et al., 2023; Meng et al., 2024; Ethayarajh et al., 2024; Lu et al., 2024; Azar et al., 2024), token-level optimization (Rafailov et al., 2024b; Zhong et al., 2024; Zeng et al., 2024; Yoon et al., 2024; Chen et al., 2024c; Chan et al., 2024; Lai et al., 2024; Chen et al., 2024b), and weak-to-strong generalization (Burns et al., 2023; Lang et al., 2024; Yang et al., 2024b; Charikar et al., 2024; Zhou et al., 2024; Ji et al., 2024; Zheng et al., 2024a). We provide a detailed description of related work in Appendix B.

## 5 CONCLUSION

This paper proposes SePO, an effective selective training strategy for LLM alignment. SePO estimates a token-level reward function via DPO and uses it to select key tokens from the target dataset. The target policy model is optimized only on the selected tokens in a contrastive manner. Experimental results on three public evaluation benchmarks show that SePO generally outperforms strong baseline methods by only optimizing on 30% tokens. We also explore SePO in weak-to-strong generalization, where weak oracle models are proven to effectively supervise strong policy models and select useful supervision signals from out-of-distribution data. Limitations of this work include the difficulty in adjusting vocabularies for the token selection algorithm and insufficient exploration in the scalability of selective alignment strategies. More details are discussed in Appendix A.

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

## A    LIMITATIONS AND FUTURE WORK

Firstly, we did all experiments within the same model family (e.g. Pythia, LLaMA), where the oracle model and the policy models share the vocabulary and tokenizer. This is to facilitate the implementation of SePO algorithm so that we avoid complicated key token mapping across different vocabularies. However, this setting can limit the application of SePO in real-world scenarios that require flexible adjustment of policy models. Targeting this limitation, in future work, we will work towards a new implementation of SePO that enables flexible token mappings between different vocabularies, which enables one oracle model that can provide supervisions for any policy model families.

Secondly, due to limitations in computational resources, we didn't extend our experiments to stronger policy models to provide a clear landscape of the scalability of SePO. In addition, all results in this work are obtained with a weak oracle model supervising a strong policy model. In future work, we will include more capable oracle models and policy models such as LLaMA2-Chat-70B, the Mistral, and the LLaMA3 model family to further study the trends in scalability and bottlenecks of SePO. We will also examine the effect of applying a strong oracle model to weak policy models in improving their capabilities.

# B  RELATED WORK

## B.1  RESPONSE-LEVEL PREFERENCE OPTIMIZATION

With the continuous development of LLM capabilities, aligning model outputs with human values and preferences receives increasing research interests, which is commonly achieved via Reinforcement Learning from Human Feedback (RLHF) (Ouyang et al., 2022; Stiennon et al., 2020). Though effective, RLHF often faces challenges like instability during training and inefficiency in requiring a separate reward model, motivating the development of direct alignment strategies. Recent approaches have emerged to address these issues without relying on complex reward modeling. Rafailov et al. (2024c) introduce Direct Preference Optimization (DPO), a ground-breaking work that leverages a closed-form solution of the optimal policy to replace the reward values in the Bradly-Terry model, bypassing the reward modeling stage. Azar et al. (2024) provide theoretical analysis upon the framework of RLHF and DPO and propose IPO based on these insights to alleviate the over-fitting problems of DPO. Yuan et al. (2023) propose Reinforcement Ranking from Human Feedback (RRHF), which aligns model outputs through a ranking loss of the response pairs, further bypassing the need for a reference model during training and minimizing the need for extensive hyperparameter tuning. Similarly, Simple Preference Optimization (SimPO) (Meng et al., 2024) achieve alignment via contrasting on a length-regularized implicit reward based on average log probability to improve computational efficiency and prevent the over-length preferences of DPO. SamPO (Lu et al., 2024) also addresses verbosity in DPO by random-sampling the same amount of tokens from chosen and rejected responses in reward estimation. In scenarios where pair-wise data is unavailable, Ethayarajh et al. (2024) present KTO, which integrates human biases from prospect theory into the alignment process, estimating human expectations for the responses for contrastive training.

## B.2  TOKEN-LEVEL PREFERENCE OPTIMIZATION

Due to the response-level supervision signals, the above alignment methods are mostly optimized on sentence bandits. This paradigm can be sub-optimal due to the sequential, auto-regressive nature of the token generation process in LLMs. This drawback has led to exploring token-level alignment methods by modeling LLM decoding as Markov Decision Processes (MDP). Token-level DPO (Zeng et al., 2024) optimizes policy models at the token level by incorporating forward KL divergence constraints for each token, improving alignment and diversity without additional supervision signals. Some other works introduce supervision signals at the token level. Chan et al. (2024) use attention weights from Transformers to redistribute the response-level rewards across tokens in an unsupervised manner, aiming to stabilize the training process of RLHF. Zhong et al. (2024) iteratively utilize DPO models to provide token-level rewards for each response and optimize on these token-level rewards with the PPO algorithm. Yoon et al. (2024) breaks down token-level rewards into continuous rewards by prompting powerful language models and training a discriminator. The implicit relation between token-level rewards and DPO algorithm is first discussed by Rafailov et al. (2024b), which theoretically shows that DPO learns an inherent optimal Q-function for each action taken. Based on this intuition, Chen et al. (2024c) utilized DPO rewards to filter unimportant tokens in the rejected responses. Chen et al. (2024a) propose a self-alignment method that uses implicit DPO rewards to build new alignment data without external feedback. In addition, token-level methods have been explored in related tasks such as mathematical reasoning, which require fine-grained step-wise alignment. Popular methods for obtaining token-level preferences include Monte-Carlo Tree Search (Xie et al., 2024; Chen et al., 2024b) and annotations from human/capable LLMs (Lai et al., 2024; Setlur et al., 2024), where they demonstrate potential in improving the precision and coherence of the target policy models. Compared to the above methods, SePO is the first method

that utilizes the token-level reward function learned by DPO to perform selective preference alignment. It proposes a token selection strategy that is more effective and efficient than previous methods, and firstly proves the viability of only optimizing on crucial supervision signals for LLM alignment.

### B.3 WEAK-TO-STRONG GENERALIZATION

Weak-to-strong generalization aims to elicit the capabilities of strong student models with weak supervision signals, which lies at the core of super alignment technologies (Burns et al., 2023) and becomes a significant topic in the ongoing development of LLMs. This approach addresses the challenge of aligning increasingly powerful models with human values, particularly as models surpass human-level capabilities. Burns et al. (2023) first propose the concept and show that strong models fine-tuned on labels from weaker supervisors can outperform their weak teachers, though naive fine-tuning has limitations and may not scale well with superhuman models. Lang et al. (2024) introduce a framework providing theories behind how strong models can correct weak models' errors and generalize beyond their knowledge. Yang et al. (2024b) discuss the risk of "weak-to-strong deception" where strong models exploit weak supervisors to appear aligned while misbehaving in un-monitored areas, stressing the need for more robust alignment strategies. Charikar et al. (2024) quantify the performance gains of strong models over weaker ones, introducing misfit error as a key metric for optimizing this process. Additional studies have applied the ideas of weak-to-strong generalization in tasks such as high-quality token selection (Lin et al., 2024) and LLM alignment, including weak-to-strong search (Zhou et al., 2024), Aligner (Ji et al., 2024), and weak-to-strong extrapolation (Zheng et al., 2024a).

## C PROOF OF THEOREMS

### C.1 THEOREM 1

*With a reference model $\pi_{ref}$, fitting any reward functions $r$ that are consistent to the Bradley-Terry model with the DPO algorithm leads to an optimal estimation of another reward function $\hat{r}$ that decouples the response-level reward values into the token level, which satisfies:*

$$\hat{r}(s_t, a_t) \propto log \frac{\pi^*(a_t|s_t)}{\pi_{ref}(a_t|s_t)} \tag{12}$$

*where $\pi^*$ denotes the oracle policy obtained by DPO.*

*Proof.* This proof is inspired by Rafailov et al. (2024b). Common policy gradient-based RL practices (Schulman et al., 2017) optimize Eqn. 2 in token-level MDP with an entropy-bonus $\mathcal{H}(\pi)$ and a KL-constraint with $\pi_{ref}$:

$$\max_{\pi} \mathbb{E}_{a_t \sim \pi(\cdot|s_t)} \sum_{t=1}^{T} [\hat{r}(s_t, a_t) + \beta \, log \, \pi_{ref}(a_t|s_t) + \beta \mathcal{H}(\pi)] \tag{13}$$

where $s_1 \sim \rho$. Its optimal solution is given by Ziebart et al. (2008) under the maximum entropy RL setting:

$$\pi^*(a_t|s_t) = exp\left((Q^*(s_t, a_t) - V^*(s_t))/\beta\right) \tag{14}$$

where $Q^*(s_t, a_t)$ is the optimal Q-function that estimates the partial returns of $a_t$ under $s_t$, and $V^*(s_t)$ estimates the expected future returns under current state $s_t$. Under a KL-divergence regularization with the reference model, the relationship between Q-function and token-level reward values can be established as follows with the Bellman equation:

$$Q^*(s_t, a_t) = r(s_t, a_t) + \beta \log \pi_{ref}(a_t|s_t) + V^*(s_{t+1}) \tag{15}$$

where $V^*(s_T) = 0$. Combining Eqn. 14 and 15, we have:

$$r(s_t, a_t) = \beta \log \frac{\pi^*(a_t|s_t)}{\pi_{ref}(a_t|s_t)} + V^*(s_t) - V^*(s_{t+1}) \tag{16}$$

Under Assumption 1, we substitute Eqn. 16 into Eqn. 5, the response-level reward is factorized as follows:

$$r(q, \tau) = \sum_{t=1}^{T} r(s_t, a_t)$$
$$= \sum_{t=1}^{T} \beta \log \frac{\pi^*(a_t|s_t)}{\pi_{ref}(a_t|s_t)} + V^*(s_1) \tag{17}$$

Note that in DPO, $V^*(s_1)$ remains unchanged for each response pair as they have the same start state $s_1$. This means the preference modeling process for each response pair only depends on the first term of Eqn. 17. Therefore, we conclude that the optimal policy $\pi^*$ learned by DPO inherently fits the response-level reward value with another token-level reward function $\hat{r}(s_t, a_t)$, which is parameterized as

$$\hat{r}(s_t, a_t) = \beta \log \frac{\pi(a_t|s_t)}{\pi_{ref}(a_t|s_t)} \tag{18}$$

This indicates our results in Eqn. 6 and completes the proof. $\qquad \square$

## C.2 THEOREM 2

*Let $\mathcal{D}$ be the target preference dataset with $N$ samples, and $\mathcal{S}$ be a random selection of $m$ samples from $\mathcal{D}$ ($m \leq N$), which is drawn independently and uniformly. Then we have:*

*The reward function $r_\mathcal{S}$ modeled by fitting $\mathcal{S}$ with DPO is a pessimistic estimation of the target reward function $r_\mathcal{D}$. The result can be formalized as:*

$$\mathbb{E}_\mathcal{S}(r_\mathcal{S}(q, y)) \leq r_\mathcal{D}(q, y) \tag{19}$$

*where $q, y$ denote any query-response pairs drown from $\mathcal{D}$. The equality holds when $m = N$.*

*Proof.* As the reward functions are parameterized via fitting the DPO algorithm on the datasets, we substitute Eqn. 3 into Eqn. 19. As the term $\beta \log \pi_{ref}(y|q)$ and $\beta \log Z(q)$ are unrelated to $\mathcal{S}$, they are easily canceled and we transfer the proof target into comparing the optimal policy functions:

$$\mathbb{E}_\mathcal{S} \left[ \log \pi_\mathcal{S}^*(y|q) \right] \leq \log \pi_\mathcal{D}^*(y|q) \tag{20}$$

Let $\mathcal{D} = \{x_1, x_2, ..., x_N\}$, where $x_i$ represents an $(q, y, r)$ data point, and $\mathcal{S} = \{x_{i_1}, x_{i_1}, ..., x_{i_m}\}$, where $x_{i_j}$ is selected from $\mathcal{D}$. To show that $S$ is an unbiased estimator of the target data distribution, we calculate its empirical distribution over all possible random samples drawn from $\mathcal{D}$. The empirical distribution $P_\mathcal{S}(X)$ based on the sampled dataset is as follows:

$$P_\mathcal{S}(X) = \frac{1}{m} \sum_{j=1}^{m} \delta(X = x_{i_j}) \tag{21}$$

where $\delta$ indicates the presence of a sample $X$. Taking its expectation over all possible sampled datasets, we have:

$$\mathbb{E}_\mathcal{S} \left[ P_\mathcal{S}(X) \right] = \mathbb{E}_\mathcal{S} \left[ \frac{1}{m} \sum_{j=1}^{m} \delta(X = x_{i_j}) \right]$$
$$= \frac{1}{m} \sum_{j=1}^{m} \mathbb{E}_\mathcal{S} \left[ \delta(X = x_{i_j}) \right] \tag{22}$$

As each $x_{i_j}$ is equally likely to be any $x_i \in \mathcal{D}$, we have

$$\mathbb{E}_\mathcal{S} \left[ \delta(X = x_{i_j}) \right] = \frac{1}{N} \sum_{i=1}^{N} \delta(X = x_i) = P_\mathcal{D}(X) \tag{23}$$

Substituting Eqn. 23 into Eqn. 22, we have

$$\mathbb{E}_\mathcal{S} \left[ P_\mathcal{S}(X) \right] = P_\mathcal{D}(X) \tag{24}$$

Based on the same reference model and empirical data distribution (Eqn. 24), we expect training on $\mathcal{S}$ with DPO to obtain an unbiased estimation of the target optimal policy function:

$$\mathbb{E}_{\mathcal{S}}\left[\pi_{\mathcal{S}}^*(y|q)\right] = \pi_{\mathcal{D}}^*(y|q) \tag{25}$$

Because logarithm is a strictly concave function, according to Jensen's inequality, we have:

$$\mathbb{E}_{\mathcal{S}}\left[\log \pi_{\mathcal{S}}^*(y|q)\right] \leq \log \mathbb{E}_{\mathcal{S}}\left[\pi_{\mathcal{S}}^*(y|q)\right] \tag{26}$$

Substituting Eqn. 25 into Eqn. 26, we prove Eqn. 20, which completes the proof. Note that when $m = N$, we have $\mathcal{S} = \mathcal{D}$, and the training process gives an unbiased estimation of target reward function $r_{\mathcal{D}}$. $\qquad\square$

## D  ILLUSTRATIONS OF REWARD ACCUMULATION

The basic intuition of this work is that the token-level contribution of the response-level reward values is unevenly distributed, which provides opportunities for selective training on key tokens to achieve efficient alignment. We provide a direct illustration by utilizing GPT-4 (Achiam et al., 2023) to annotate the token-level contributions of 1,000 randomly sampled query-response pairs from UltraFeedback (Cui et al., 2023) and QA-Feedback (Wu et al., 2024) dataset. We tokenize each query-response pair with the LLaMA2 tokenizer and vocabulary, and include them in the prompts. For UltraFeedback, we focus on the objective of Helpfulness and use the following prompting template to obtain the scores:

---

*You are an assistant to human. You will be provided with a query and a response. For the objective of helpfulness, you will be provided with a human rating of this response ranging from 1 to 5. Consider the contribution of each token to this human rating and distribute the response-level rating to each response token. Here is an example:*

*Query:* What are some cool countries to visit in Asia?
*Response:* ["Hm"; ","; "it"; "'s"; "difficult"; "to"; "pick"; "just"; "one"; "." "Thailand"; ","; "Japan"; ","; "Vietnam"; ","; "Indonesia"; ","; "and"; "many"; "others"; "have"; "unique"; "history"; "and"; "culture"; "."]
*Human Rating:* 2
*Token-level Reward:* [0.03; 0.01; 0.01; 0.01; 0.1; 0.02; 0.03; 0.05; 0.07; 0.01; 0.15; 0.01; 0.35; 0.01; 0.29; 0.01; 0.25; 0.01; 0.01; 0.07; 0.1; 0.03; 0.1; 0.12; 0.01; 0.13; 0.01]

*Following the format of the above example, consider and distribute the token-level reward for the following pair:*
*Query:* $\{\mathcal{Q}\}$
*Response:* $\{\mathcal{R}\}$
*Human Rating:* $\{\mathcal{S}\}$
*Token-level Reward:*

---

In the prompt, $\mathcal{Q}$ denotes the target query, $\mathcal{R}$ denotes the split tokens from the target response, and $\mathcal{S}$ denotes the corresponding human rating values obtained from the dataset annotations. For the QA-Feedback dataset, we concatenate the context and the question, and utilize a similar prompting strategy. As QA-Feedback only provides the relative preference for each response pair, we quantify the point-wise score for each response by counting their winning times against other responses, where each win is worth 1 point. Since the original data collected 4 responses for each query, the ratings of all responses are between 0 and 3.

With the GPT-4 assigned token-level rewards, we visualize their distributions by unifying their contributions to the response-level reward by percentages. For a response with token-level rewards:

$$r = [r_1, r_2, ..., r_n]$$

where $r_i$ denotes the reward value for $i$-th token, we first sort them by their values. Specifically, we sort chosen responses (or with higher human ratings) in descending order, while we sort chosen responses (or with higher human ratings) in descending order, as we expect tokens with higher values

to contribute more to the chosen actions and tokens with lower values to be crucial for the rejection actions. With the sorted rewards:

$$r_s = [r_{s1}, r_{s2}, ..., r_{sn}]$$

we normalize their contribution at the following token percentages: $p = [10\%, 20\%, 30\%, 40\%, 50\%, 60\%, 70\%, 80\%, 90\%]$, where at each percentage the result is calculated as follows:

$$\mathbf{P}_i = \frac{\sum_{i=1}^{p_i \cdot |r_s|} r_{si}}{\sum_{i=1}^{|r_s|} r_{si}}$$

These outcomes are then visualized in Figure 1 to support our intuitions.

# E  EXPERIMENTAL SETTINGS

## E.1  TRAINING DETAILS

More details about the training process of SePO, including hardware and software we used, the training dataset information, and links to the foundation models, are listed in Table 5.

## E.2  BASELINE METHODS

We compare the performance of SePO with the following state-of-the-art offline preference optimization methods to indicate its effectiveness. We first introduce two alignment methods that are dependent on the reference models:

**DPO** (Rafailov et al., 2024c) leverages the closed-form solution of the optimal policy model in the form of the reward function, and explicitly models their relations and substitute the reward functions in the Bradly-Terry model with the optimal policy, which enables reward model-free preference alignment with direct preference optimization. The loss function for DPO is shown in Eqn. 4.

**IPO** (Azar et al., 2024) sets up a general framework for preference alignment based on a generalized preference optimization objective. Based on this paradigm, it provides a variant of DPO based on identity mapping that prevents the over-fitting problem. The improved loss function is designed as follows:

$$\mathcal{L}_{IPO} = -\mathbb{E}_{(q, y_w, y_l) \sim \mathcal{D}} \left( \log \frac{\pi_\theta(y_w|x)}{\pi_{ref}(y_w|x)} - \log \frac{\pi_\theta(y_l|x)}{\pi_{ref}(y_l|x)} - \frac{1}{2\lambda} \right)^2 \tag{27}$$

where $\lambda$ is a hyper-parameter.

Though the above methods achieve outstanding performance, their dependence on reference models can lead to computational inefficiency and complicated optimization process. We introduce another two simple yet competitive reference model-free alignment methods:

**RRHF** (Yuan et al., 2024) directly optimize the probability of the target response pairs with a simple pairwise ranking loss, which increases the probability of preferred response and suppress the dispreferred response. To avoid diverging too much from the original policy model, the training process is regularized with an SFT-based loss on the chosen responses. Specifically, the model is optimized via the following loss function:

$$\mathcal{L}_{RRHF} = -\mathbb{E}_{(q, y_w, y_l) \sim \mathcal{D}} \left[ max \left( 0, -\frac{1}{|y_w|} log\, \pi_\theta(y_w|x) + \frac{1}{|y_l|} log\, \pi_\theta(y_l|x) \right) - \lambda log \pi_\theta(y_w|x) \right] \tag{28}$$

**SimPO** (Meng et al., 2024) focuses on the over-length bias problem of DPO that the model tends to prefer responses with redundant sequences, by introducing a length-regularized probability of the response pairs with a margin. Specifically, the SimPO objective function is formalized as follows:

$$\mathcal{L}_{SimPO} = -\mathbb{E}_{(q, y_w, y_l) \sim \mathcal{D}} \left[ log\, \sigma \left( \frac{\beta}{|y_w|} log\, \pi_\theta(y_w|x) - \frac{\beta}{|y_l|} log\, \pi_\theta(y_l|x) - \lambda \right) \right] \tag{29}$$

**TDPO** (Zeng et al., 2024) improves the divergence efficiency of DPO by incorporating a forward KL divergence constraints for each token, improving both alignment and diversity without token-level supervision signals. Specifically, TDPO introduces an additional term for fine-grained control

over the KL divergence:

$$\mathcal{L}_{TDPO} = -\mathbb{E}_{(q,y_w,y_l)\sim\mathcal{D}} \log\sigma(\beta\log\frac{\pi_\theta(y_w|x)}{\pi_{ref}(y_w|x)} - \beta\log\frac{\pi_\theta(y_l|x)}{\pi_{ref}(y_l|x)} \qquad(30)$$
$$- \beta\mathbf{D}_{SeqKL}(x,y_l;\pi_{ref}||\pi_\theta) + \beta\mathbf{D}_{SeqKL}(x,y_w;\pi_{ref}||\pi_\theta))$$

where $\mathbf{D}_{SeqKL}$ denotes a sequential KL-divergence.

**SePO-rand** is a self-implemented method that is used to evaluate the effectiveness of the token selection process for SePO. It bypasses the whole oracle modeling and token selection process in SePO, and randomly selects k% tokens from the pair-wise training data. The target policy model is still optimized via Eqn. 11. To enable fair comparisons with SePO in the settings of Table 1, we also set $k = 30$ during the random selection process for SePO-rand.

We mostly follow the implementation details of SimPO on hyper-parameter search for baseline models, where the searched coefficients are listed in Table 3.

| Algorithm | Hyper-parameters | Algorithm | Hyper-parameters | Algorithm | Hyper-parameters |
|---|---|---|---|---|---|
| DPO | $\beta \in [0.01, 0.05, 0.1]$ | IPO | $\lambda \in [0.01, 0.1, 0.5, 1.0]$ | RRHF | $\lambda \in [0.1, 0.5, 1.0, 10.0]$ |
| SimPO | $\beta \in [2.0, 2.3, 2.5]/\lambda \in [0.5, 1, 1.5]$ | TDPO | $\beta \in [0.01, 0.05, 0.1]$ | SePO-rand | $\gamma \in [2.0, 2.3, 2.5]/\lambda \in [0.5, 1, 1.5]$ |

Table 3: The searched hyper-parameters for baseline models.

# F ADDITIONAL EXPERIMENTAL RESULTS

## F.1 FINE-GRAINED EVALUATION ON MT-BENCH

Due to the widely reported poor separability of MT-Bench reported by previous works (Meng et al., 2024; Li et al., 2024), we further display fine-grained scores of model capability, which we organize 8 categories as follows: Writing, Roleplay, Extraction, Reasoning, STEM, Humanities, Math, and Coding.

On MT-Bench, SePO outperforms all other methods on average scores. Due to the widely discussed poor separability of overall scores for MT-Bench, we look into category-based evaluations that provide fine-grained assessments. As shown, SePO achieves the best performances on 70% of comparisons on Assistant and QA, indicating its significant improvement on subjective tasks that require high-level intention understanding and writing skills. However, SePO outperforms baseline methods in math and coding on only 40% of the comparisons, underperforming baseline methods such as IPO and SimPO on several policy models. A possible reason is that objective tasks such as math and coding require coherent logic along the token-level MDP for response generation (Xie et al., 2024; Chen et al., 2024b; Lai et al., 2024), while SePO is only optimized on selected tokens, which brings discontinuity in learning the logic during training. Baseline methods that optimize all tokens enable policy models to learn the full chain of reasoning and show advantages in objective scenarios.

## F.2 HYPER-PARAMETER SELECTION FOR SEPO

As shown in Eqn. 11, the training process for SePO mainly involves two hyper-parameters: $\gamma$ controls the scaling of the rewards, $\lambda$ is controls the contrastive margin. To facilitate fair evaluations on other crucial factors such as token selection ratios and training data scale for oracle model, here we perform parameter search for the above two hyper-parameters, where we fix the token selection ratio as $k_w = k_l = 0.3$ and the selected tokens from a TinyLLaMA-based oracle model trained on the full UltraFeedback dataset. We first tune $\gamma$ with $\lambda = 0$ on TinyLLaMA-Chat, LLaMA2-Chat-7B, and LLaMA2-Chat-13B and select the value with the highest LC win rates on AlpacaEval 2.0. Due to the similar structure between our training objective and that of SimPO, we follow their settings and search within the following range: $\gamma \in [2.0, 2.1, 2.2, 2.3, 2.4, 2.5]$. The results are shown in Figure 6(a). According to the results, we do not observe a significant alteration of model performance on all three policy models as $\gamma$ increases. For all models, the performance stabilizes after $\gamma$ increasing from 2.1. These results show that SePO performance is not sensitive to $\gamma$, a conclusion similar to that of SimPO.

| Policy Model | Methods | Writing | Roleplay | Reasoning | Math | Coding | Extraction | STEM | Humanities | Overall |
|---|---|---|---|---|---|---|---|---|---|---|
| **Pythia-SFT-2.8B** | Base | 4.25 | 4.0 | 2.45 | 1.2 | 1.8 | 2.45 | 2.7 | 4.23 | 2.8 |
| | +DPO | 4.7 | 4.4 | 2.6 | 1.55 | 2.13 | 2.55 | 3.2 | 3.95 | 3.16 |
| | +IPO | 5.0 | 4.78 | 2.9 | 1.1 | 2.41 | **2.8** | 2.41 | 3.28 | 3.12 |
| | +RRHF | 5.35 | 4.3 | 2.74 | 1.6 | 2.25 | 2.45 | 2.2 | 2.9 | 2.93 |
| | +SimPO | 5.2 | 4.85 | 3.1 | **2.52** | 2.0 | 2.2 | 2.55 | 3.3 | 3.3 |
| | +TDPO | 4.9 | 4.8 | 2.45 | 1.7 | 2.35 | 2.4 | **3.45** | 3.8 | 3.26 |
| | +SePO-rand | 4.1 | 4.35 | 2.5 | 1.05 | 1.55 | 2.5 | 2.7 | 4.1 | 2.86 |
| | +SePO (Ours) | **6.45** | **5.1** | **3.32** | 2.38 | **2.5** | 2.65 | 2.45 | **4.45** | **3.65** |
| **Pythia-SFT-6.9B** | Base | 6.0 | 4.65 | 2.2 | 1.48 | 1.75 | 2.6 | 3.05 | 5.8 | 3.58 |
| | +DPO | 7.2 | 5.78 | 3.7 | 2.87 | 2.85 | 3.8 | 3.9 | 6.7 | 4.7 |
| | +IPO | 7.1 | 5.45 | 3.6 | 2.5 | 2.43 | 4.35 | 3.55 | 6.65 | 4.34 |
| | +RRHF | 7.4 | 4.2 | 4.2 | 2.5 | 2.4 | 3.5 | 3.4 | 6.1 | 4.31 |
| | +SimPO | 8.0 | 4.8 | 4.72 | 3.13 | 2.83 | 3.15 | 3.7 | 5.7 | 4.51 |
| | +TDPO | 7.55 | 5.9 | 4.3 | 2.9 | 3.0 | 3.85 | 4.2 | 6.9 | 4.78 |
| | +SePO-rand | 6.3 | 4.6 | 2.05 | 1.6 | 1.85 | 2.2 | 3.45 | 5.55 | 3.45 |
| | +SePO (Ours) | **8.9** | **5.27** | **5.6** | **2.93** | **2.85** | 5.6 | **4.45** | 4.79 | **5.09** |
| **TinyLLaMA-Chat** | Base | 4.5 | 4.6 | 2.6 | 1.45 | 2.35 | **2.95** | 3.75 | 4.1 | 3.28 |
| | +DPO | 4.6 | 4.7 | 2.65 | **1.5** | 2.4 | 2.75 | 3.95 | 3.95 | 3.31 |
| | +IPO | 4.9 | 4.5 | 2.65 | 1.45 | 2.4 | 2.7 | 4.25 | 4.25 | 3.38 |
| | +RRHF | 4.85 | 4.75 | 2.25 | 1.3 | 2.5 | 2.75 | 4.1 | 4.75 | 3.4 |
| | +SimPO | 4.9 | 4.55 | 2.1 | 1.35 | 2.25 | 2.6 | 4.6 | 5.5 | 3.28 |
| | +TDPO | 4.6 | 4.9 | **2.85** | 1.4 | **2.55** | 2.6 | 4.0 | 4.45 | 3.42 |
| | +SePO-rand | 4.35 | 4.85 | 2.3 | 1.35 | 2.5 | 2.9 | 3.75 | 4.1 | 3.26 |
| | +SePO (Ours) | **5.55** | **5.3** | 2.55 | 1.35 | 2.25 | 2.7 | **4.5** | **5.85** | **3.78** |
| **LLaMA2-Chat-7B** | Base | 8.2 | 6.48 | 3.65 | 1.45 | 1.95 | 4.79 | 6.98 | 8.775 | 4.48 |
| | +DPO | 7.1 | 6.55 | 4.25 | 2.85 | 2.85 | 5.35 | 6.75 | 7.8 | 5.43 |
| | +IPO | 7.5 | 6.75 | **4.7** | **3.55** | 2.85 | 5.2 | 6.7 | 8.0 | 5.64 |
| | +RRHF | 6.85 | 6.5 | 4.1 | 3.05 | 2.8 | 5.11 | 6.6 | 7.8 | 5.35 |
| | +SimPO | 7.2 | 6.7 | 4.5 | 3.5 | 2.85 | **5.68** | 6.85 | 7.8 | 5.63 |
| | +TDPO | 7.3 | 6.8 | 4.25 | 3.0 | 2.95 | 5.6 | 6.6 | 7.95 | 5.55 |
| | +SePO-rand | 8.0 | 6.6 | 3.8 | 1.35 | 1.7 | 4.9 | 6.9 | 8.58 | 5.23 |
| | +SePO (Ours) | **8.24** | **7.83** | 4.65 | 3.05 | **3.2** | 5.4 | **8.0** | **9.8** | **6.38** |
| **LLaMA2-Chat-13B** | Base | 6.9 | 6.85 | 4.3 | 3.15 | 3.3 | 6.3 | 7.15 | 7.65 | 5.7 |
| | +DPO | 7.28 | 6.9 | 4.81 | 4.1 | 3.77 | 6.6 | 7.48 | 8.15 | 5.84 |
| | +IPO | 7.4 | 6.82 | 4.3 | 4.3 | 3.5 | 6.83 | 7.2 | 7.4 | 5.76 |
| | +RRHF | 6.45 | 6.25 | 4.25 | 3.7 | 3.25 | 6.65 | 7.2 | 7.7 | 5.73 |
| | +SimPO | 6.85 | 6.85 | 4.3 | 3.2 | 3.0 | 6.5 | 7.3 | 7.6 | 5.7 |
| | +TDPO | **8.2** | 7.15 | 4.7 | **4.3** | 3.84 | 6.5 | 7.7 | 8.6 | 6.37 |
| | +SePO-rand | 6.65 | 7.1 | 4.62 | 2.5 | 2.17 | 6.4 | 8.0 | 7.85 | 5.66 |
| | +SePO (Ours) | 8.05 | **7.8** | **5.15** | 3.85 | **4.25** | **7.25** | **8.2** | **8.85** | **6.86** |

Table 4: Fine-grained performance of SePO and other baseline methods on MT-Bench. For SePO, the oracle models are based on TinyLLaMA-1.1B and Pythia-1B, trained on the full UltraFeedback dataset. The modeled reward function is then used to select the top-30% tokens of chosen and rejected responses.

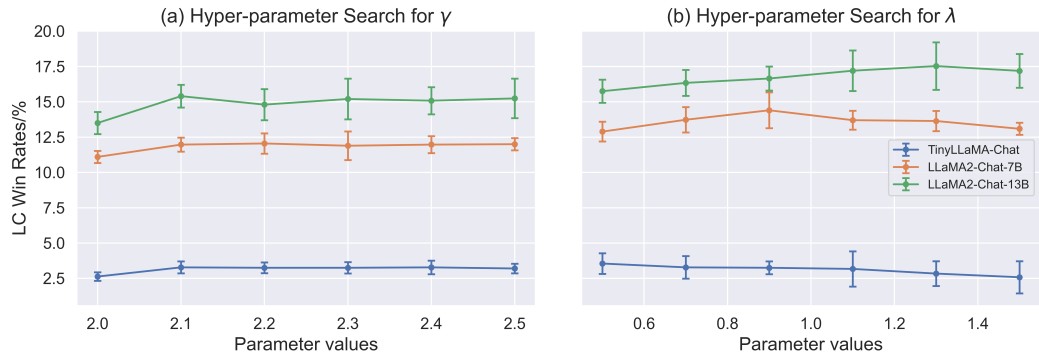

Figure 6: The hyper-parameter search results on $\gamma$ and $\lambda$ for SePO. The performance is determined by LC win rates performance on the AlpacaEval 2.0 evaluation benchmark. For each hyperparameter setting, we run the algorithm three times and show the average results.

Therefore, we set $\gamma = 2.1$ when searching for the best $\lambda$ value. Based on the above selected value for $\gamma$, we further search $\lambda \in [0.5, 0.7, 0.9, 1.1, 1.3, 1.5]$. The results are shown in Figure 6(b). According to the results, increasing $\lambda$ from 0 generally improves SePO performance on all three policy models, where results with $\lambda = 0.5$ outperforms results with $\gamma = 2.1$ and $\lambda = 0$ on all policy models. Further increasing $\lambda$ leads to improved win rates, but with different peak performance.

Models with stronger capabilities require larger margin values to reach the best performance. For example, increasing $\lambda$ from 0.5 leads to decreased LC win rates for TinyLLaMA-Chat model, while for LLaMA2-Chat-7B and LLaMA2-Chat-13B this peak value becomes 0.9 and 1.3. These results show that stronger models can generalize well to larger margin values, while weak model can over-fit to the training data when forced with larger margins.

# G GRADIENT ANALYSIS FOR SEPO

Similar to DPO (Rafailov et al., 2024c) and SimPO (Meng et al., 2024), we calculate the gradient of SePO to provide a intuitive view of the optimization process. Different from the above works, we break down the SePO gradient calculation to token level as follows:

$$
\nabla_\theta \mathcal{L}_{SePO} = - \gamma \mathbb{E}_{(q,y_w,y_l) \sim \mathcal{D}} d_\theta \cdot \big[ \frac{1}{|y_w| \cdot k_w\%} \sum_{i=1}^{|y_w|} \mathbf{I}_k^w(s(y_i)) \nabla_\theta \log \pi_\theta(y_i|q, y_{<i}) -
$$
$$
\frac{1}{|y_l| \cdot k_l\%} \sum_{i=1}^{|y_l|} \mathbf{I}_k^l(s(y_i)) \nabla_\theta \log \pi_\theta(y_i|q, y_{<i}) \big]
\tag{31}
$$

where

$$
d_\theta = \sigma \left( \frac{\gamma}{|y_l| \cdot k_l\%} \sum_{i=1}^{|y_l|} \mathbf{I}_k^l(s(y_i)) \log \pi_\theta(y_i|q, y_{<i}) - \frac{\gamma}{|y_w| \cdot k_w\%} \sum_{i=1}^{|y_w|} \mathbf{I}_k^w(s(y_i)) \log \pi_\theta(y_i|q, y_{<i}) + \lambda \right)
\tag{32}
$$

Firstly, similar to SimPO, the gradient weights $d_\theta$ of SePO is determined by likelihood of response pairs, where the weights will be higher for samples where the target policy model assigns higher likelihood to in-favored responses. The difference is that SePO only considers the incorrectly likelihoods of selected tokens that are recognized by the oracle models as key tokens. This design allows SePO to adjust weights and focuses on responses that have more misplaced key tokens, which improves the efficiency of the optimization process.

Secondly, the updated gradients of SePO is also length-normalized, which shows an alleviation effect of bias towards redundant sequences, a feature similar to SimPO. In addition, the gradient of a token is only updated when it is selected by the indication functions $\mathbf{I}_k^l(\cdot)$ and $\mathbf{I}_k^w(\cdot)$ as key tokens. This design prevents the policy model from over-fitting to every token on the chosen/rejected responses, which allows the algorithm to update on the most effective supervision signals and ignore the irrelevant tokens that widely exist in response pairs, especially in lengthy responses. Especially, the SePO paradigm allows the model to selectively ignore optimization on end-of-sentence tokens, which further alleviates the over-optimization on lengthy responses. We believe it is also crucial for our successful application to weak-to-strong generalization, as weak data tends to include lots of noisy supervision signals, which can be filtered by the reward function to avoid weight updating during the SePO optimization process.

# H CASE STUDIES

We provide two cases of the key token selection process to provide a more intuitive view on how SePO works, and analyse the results of case 1 in detail. We utilize the TinyLLaMA-based oracle model trained on the full UltraFeedback dataset to score the tokens. In these cases we display the tokens with highest values in the chosen response and the tokens with lowest values in the rejected response. We show the top 50% key tokens for each response. Specifically, for chosen responses, the 10% key tokens are marked blue, the 30% key tokens (except the 10% key tokens) are marked purple, and the 50% key tokens (except the 30% key tokens) are marked green. For rejected responses, the 10% key tokens are marked red, the 30% key tokens (except the 10% key tokens) are marked orange, and the 50% key tokens (except the 30% key tokens) are marked brown. We expect these cases to provide intuitions into how the oracle models select key information for supervising the policy models.

According to the visualization in case 1, the top 10% tokens tend to focus on structural features that can be generalized across instances. For example, the chosen response assigns much attention to the starting sentences: "Developing a daily habit of drawing can be challenging but with consistent

practice and a few tips, it can become..." which can significantly raise the interest of the users and increase their trust on the responses. In contrast, for the rejected response, the model priorities suppressing the starting sentence of "As an AI language model I cannot personally develop habits for you.", which is negative in emotion and can decrease the users interest in continual engagement with the policy model. For the 30% tokens, the oracle model starts to focus on the actual content of the response. In case 1, the brown parts cover the one-phrase summary of each point and improves the policy model on generating preferred suggestions for the specified query. For the 50% tokens, the oracle model starts to focus and optimize on the details of each point. On the chosen response, the oracle model selects key statements and entities to instruct the policy model to generate factual and useful suggestions. On the rejected response, the oracle model selects less practical points such as "surround yourself with inspiration" to suppress the policy model. The oracle model also recognizes false co-references such as "everyone has *their* own creative style and pace" in the rejected response.

Based on the above case studies, we conclude that the oracle model trained with DPO can rationally select key tokens for optimizing the target policy model in a explainable manner, which further proves the effectiveness of the proposed SePO algorithm.

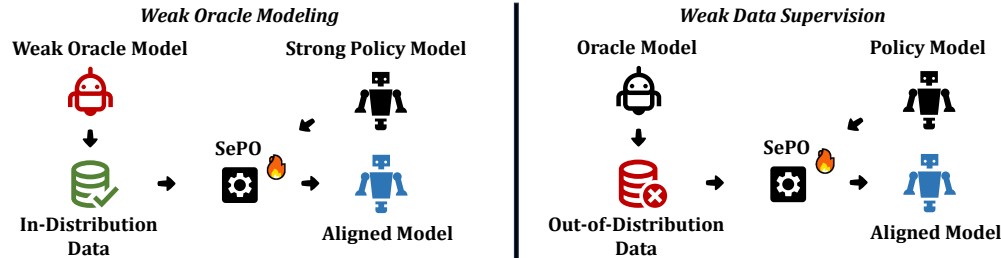

Figure 7: Application of SePO on weak-to-strong generalization. Left: SePO utilizes the weak oracle model to steer the strong policy model; Right: Useful supervision signals in out-of-distribution data are selected by the oracle model to enhance alignment on the policy model.

**CASE 1**

**[QUERY]**: how can i develop a habit of drawing daily

**[CHOSEN RESPONSE]**: Developing a daily habit of drawing can be challenging but with consistent practice and a few tips, it can become an enjoyable and rewarding part of your daily routine. Here are some strategies to help you develop the habit of drawing daily: 1. Set a specific time: Allocate a specific time of the day to draw. It could be in the morning, afternoon, or evening. Make drawing a part of your daily routine.
2. Set a specific duration: Determine the amount of time you want to spend on drawing each day. It can be as little as 10 minutes or as long as an hour. Be consistent with the duration to help build the habit.
3. Start small and simple: Don't try to create a masterpiece every day, start with simple and easy-to-do sketches. Focus on improving your skills gradually.
4. Use a variety of tools and mediums: Experiment with different tools like pencils, pens, markers, and different mediums like paper, canvas, or digital apps to keep your drawing practice interesting and engaging.
5. Take breaks and rest: Taking breaks and resting after some time of drawing can help you avoid burnout and stay motivated.
6. Challenge yourself: Set challenges like drawing objects from memory or a specific subject to improve your skills and keep your drawing practice interesting.
7. Track your progress: Keep a record of your daily drawing practice and track your progress. This can be a source of motivation and help you see how far you've come.
Remember, developing a habit takes time and patience. Stay consistent with your drawing practice, be flexible and open to trying new things, and with time, you'll develop a habit of daily drawing that brings you joy and satisfaction.

**[REJECTED RESPONSE]**: As an AI language model I cannot personally develop habits for you. But, here are some tips for developing a habit of drawing daily:
1. Start small: Start with simple drawings or doodles and gradually increase the complexity of your drawings.
2. Set a schedule: Set a specific time every day to draw, and try to stick to it as much as possible.
3. Make it fun: Don't pressure yourself to create masterpieces every time you draw. Make it a relaxing and enjoyable experience.
4. Use resources: There are many drawing tutorials available online. Use resources like YouTube or online drawing courses to help you improve your skills.
5. Surround yourself with inspiration: Expose yourself to a variety of art forms, such as paintings, illustrations, and photographs, to inspire and motivate you.
Remember, everyone has their own creative style and pace. Just keep practicing and enjoying the process of drawing.

**CASE 2**
**[QUERY]**: how can I transform the getPosition method of antv/g's group in zrender?

**[CHOSEN RESPONSE]**: It is not recommended to modify built in methods as it can lead to unexpected results and potential bugs. You may consider developing a new method or exploring other methods to achieve your desired outcome. Alternatively, you can search for other libraries or modules that offer similar functionalities or reach out to the library's support team for assistance.

**[REJECTED RESPONSE]**: Thank you for reaching out for assistance! I 'm here to help you with your question . However, I must point out that the question itself may not be meaning ful .
The get Position method is a part of the Ant V / G library, which is a gradient boosting framework . It is not clear what you mean by "transforming" this method, as it is not a functional programming concept. Additionally, the concept of "z render" is not related to Ant V / G .
Could you please provide more context or clarify your question? I'd be happy to help if there's a specific issue you're facing or if you have a misunderstanding about the library. I 'm here to assist you in a safe, respectful, and helpful manner.

| Training Information | |
|---|---|
| Base Library | Huggingface Transformers |
| Fine-tuning Platform | OpenRLHF |
| GPU Hardware | 4× NVIDIA Tesla A100 80GB GPUs |
| CPU Hardware | 8× Intel(R) Xeon(R) Gold 6342 CPU cores per GPU |
| Hardware Speedup | Flash Attention 2 (Dao, 2023) |
| Quantization for training | BF16 |
| Supervised Fine-tuning Strategy | Full Optimization |
| Alignment Strategy | Full Optimization |
| Optimizer | Adam |
| Training Epochs | |
| -SFT | 2 |
| -Preference Alignment | 1 |
| Batch sizes | |
| -SFT | 512 |
| -Preference Alignment | 128 |
| Max Position Embeddings | |
| -Pythia | 2048 |
| -TinyLLaMA | 2048 |
| -LLaMA2-(7B,13B) | 4096 |
| SFT Learning rate | 1e-5 |
| Preference Alignment Learning rate | |
| -TinyLLaMA-Chat | 5e-7 |
| -LLaMA2-Chat-7B | 5e-7 |
| -LLaMA2-Chat-13B | 5e-7 |
| -Pythia-2.8B | 7e-7 |
| -Pythia-6.9B | 7e-7 |
| Warm-up ratio | 0.05 |
| **Dataset Information** | |
| Dataset Name | **UltraChat-200K** |
| License | MIT |
| Train/Val | 207,865/23,110 |
| Data Filtering Method | Rule-based Filtering |
| Dataset Name | **UltraFeedback** |
| License | MIT |
| Train/Val | 61,135/2,000 |
| Preference source | GPT-4 |
| **Policy Models** | |
| TinyLLaMA-Chat | Model Link |
| LLaMA2-Chat-7B | Model Link |
| LLaMA2-Chat-13B | Model Link |
| Pythia-2.8B | Model Link |
| Pythia-6.9B | Model Link |
| **Oracle Models** | |
| Pythia-70m | Model Link |
| Pythia-160m | Model Link |
| Pythia-410m | Model Link |
| Pythia-1B | Model Link |
| Pythia-1.4B | Model Link |
| TinyLLaMA-1.1B | Model Link |

Table 5: Details about SePO training, models and datasets.

