# OpenReview forum: "Selective Preference Optimization via Token-Level Reward Function Estimation"
_ICLR.cc/2025/Conference — ICLR 2025 Conference Withdrawn Submission_

### Official Review · Reviewer_x9Hj · 2024-10-20

**Soundness:** 2
**Presentation:** 3
**Contribution:** 2
**Rating:** 3
**Confidence:** 4

**Summary:**

This paper propose selective preference optimization (SePO). SePO select the top-k token that dominate the final reward and train DPO on these tokens to eliminate noise and improve efficiency. Experiments shows that SePO outperforms a bunch of direct -preference learning methods.

**Strengths:**

The strengths of this paper are listed as follows

1. This paper observes that the total reward of a generated utterance is usually dominated by a few tokens. This observation is interesting and motivate the method well

2. This paper propose a token-selection-based training method, which is new and interesting to me

3. The experiments are comprehensive and results look good.

**Weaknesses:**

My concerns are listed as follows:

1. My major concern is about the token selection mechanism. The motivation behind using $\hat{r}(s_t,a_t)$ as the proxy of the reward is unclear to me. Theorem 1 only proved that $\sum \hat{r} (s_t, a_t) + V^{*}(s_1) = \sum r(s_t, a_t)$, which only guarantees that the sum of $\hat{r}$ and the sum of $r$ is the same (up to a constant). However, the value distribution of $r$ and $\hat{r}$ might still be drastically different. Therefore, the token selection based on $\hat{r}=\log \pi_{\theta} / \log \pi_{\text{ref}}$ does not make sense given the current illustration.

2. It looks like that that SePO is quite sensitive to the parameter $\gamma$. The search space of $\gamma=\{2.1, ...,2.5\}$ looks weird and it seems that there is a fluctuate of performance when $\gamma$ varies in this set. This makes a issue give that the improvement over baseline is not that significant.

3. (Minor issue): The $\propto$ in equation (6) looks like a typo

**Questions:**

See weakness section

---

### Official Review · Reviewer_mfm1 · 2024-10-31

**Soundness:** 2
**Presentation:** 2
**Contribution:** 2
**Rating:** 3
**Confidence:** 4

**Summary:**

The authors propose SePO—a method that utilizes selected tokens from an oracle model to perform preference optimization. The approach is evaluated across a wide range of models and general assistant benchmarks. The authors report that by optimizing only 30% of the tokens, they were able to surpass other methods for preference optimization.

**Strengths:**

- The idea is clear and novel.
- The reported results indicate the promise of the approach.

**Weaknesses:**

- The proof of Theorem 1, which asserts that after training a DPO, the reward function can be expressed as a decoupled reward $\hat{r}$, inherits this property (Line 810) from the assumption that the reward can be written in such a manner (Assumption 1). This raises the question of whether all reward functions can be expressed in a decoupled way. From a naive perspective, a decoupled reward is not normalized, and longer texts might have larger absolute values of reward. In my attempts to learn reward models in online settings using a decoupled approach, I found that without normalization, their accuracy dramatically reduced. Normalizing the sum of small rewards over tokens led to improvements. Therefore, I strongly feel that not all rewards can be expressed in this way. Moreover, the training objective (Equation 11) uses a normalized reward, making it unclear why Theorem 1 was presented.
- Some of the writing is ambiguous. The SePO objective (Equation 11) is hard to parse visually and would benefit from a human-understandable explanation before the equation.
- The experiments lacked exploration of the dependence of performance on the KL divergence with the reference policy. It is evident that training a policy with the SePO objective could cause it to diverge significantly. This is similar to observations in Rafailov et al. [1]. For instance, could selecting a lower $\beta$ value enable DPO or other baselines to perform better than SePO?

[1] Scaling Laws for Reward Model Overoptimization in Direct Alignment Algorithms

**Questions:**

See weaknesses

---

### Official Review · Reviewer_aUUy · 2024-11-04

**Soundness:** 3
**Presentation:** 3
**Contribution:** 3
**Rating:** 5
**Confidence:** 3

**Summary:**

This paper introduces Selective Preference Optimization (SePO) which optimizes model performance by selectively training only on key tokens with high token-level reward values using Direct Preference Optimization (DPO).

This approach significantly reduces the data requirements by focusing on 30% of the tokens, avoiding noise from less informative tokens and improving computational efficiency.

Additionally, this paper also explores weak-to-strong generalization, demonstrating that weaker oracle models can provide useful supervision for larger, more powerful policy models.

Experimental results across three benchmarks show that SePO outperforms baseline methods in alignment tasks, supporting its effectiveness and adaptability.

**Strengths:**

This paper introduces a novel token-level reward function estimator using DPO.

SePO reduces the need for extensive token optimization, demonstrating improved alignment performance while training on only 30% of tokens. This is valuable for scaling LLMs and reducing computational overhead.

The weak-to-strong generalization capability of SePO allows smaller models to supervise larger ones.

**Weaknesses:**

The experiments primarily involve relatively moderate-sized models. Testing SePO on stronger models, such as LLaMA2-Chat-70B, would provide further insights into its scalability and potential bottlenecks, especially for the weak-to-strong generalization experiment.

Compared to other methods, the improvement seems to be slight.

**Questions:**

How does SePO scale with very large policy models?

How to decide the token selection threshold more smartly on different datasets and models?

---

### Official Review · Reviewer_112v · 2024-11-04

**Soundness:** 2
**Presentation:** 3
**Contribution:** 2
**Rating:** 3
**Confidence:** 4

**Summary:**

The paper introduces Selective Preference Optimization (SePO), a novel strategy for aligning large language models (LLMs) at the token level by selectively optimizing only on key tokens. Leveraging Direct Preference Optimization (DPO), SePO identifies and optimizes high-impact tokens, reducing supervision costs while improving alignment performance on benchmarks.

**Strengths:**

SePO offers a cost-efficient alignment strategy by focusing on a subset of high-reward tokens, which reduces annotation costs.
The method demonstrates better performance on several benchmarks, surpassing existing token-level and response-level alignment methods.
SePO’s weak-to-strong generalization enables effective supervision from smaller, weaker oracle models, showing scalability across varying model sizes.

**Weaknesses:**

1. The method is limited by the requirement that oracle and policy models share the same vocabulary and tokenizer, which reduces flexibility across different model architectures.
2. The use of the DPO reward format as an automated credit assignment behaviour has been attempted by other works, and the paper's contribution is weaker as only quantifies the results of this assignment to the weights of the DPO loss.
3. Suppose the confidence given by the Oracle model is used as the gold label for the credit distribution. In that case, we do not need dpo to fit the reward distribution given by the optimal policy (https://arxiv.org/abs/2404.12358, https://arxiv.org/abs/2408.14874). Alternatively, the paper needs to discuss the error problems associated with this approximation in the method to validate the need for SePO further.
4. Performance increases are relatively minor in the experiments, and the comparison model used is GPT-4-0314 (not the current optimal model, but again, not compared to the model untrained itself to provide a more intuitive increase)

**Questions:**

1. How performance when comparing win rates to the trained models themselves.
2. The model used for the experiment is a bit old and SePO needs to prove its performance on newer open source models.
3. The theoretical part needs to be further refined, and would like to see a discussion on whether DPO credit assignments need to be constructed through token-level weighted training, and that there should exist better ideas to take advantage of this feature of the DPO reward format.

---

### Note · Authors · 2024-12-06

I have read and agree with the venue's withdrawal policy on behalf of myself and my co-authors.